# Extensive substrate recognition by the streptococcal antibody-degrading enzymes IdeS and EndoS

Abigail S. L. Sudol [1], John Butler[1], Dylan P. Ivory [1], Ivo Tews [1] & Max Crispin [1]

Enzymatic cleavage of IgG antibodies is a common strategy used by pathogenic bacteria to ablate immune effector function. The *Streptococcus pyogenes* bacterium secretes the protease IdeS and the glycosidase EndoS, which specifically catalyse cleavage and deglycosylation of human IgG, respectively. IdeS has received clinical approval for kidney transplantation in hypersensitised individuals, while EndoS has found application in engineering antibody glycosylation. We present crystal structures of both enzymes in complex with their IgG1 Fc substrate, which was achieved using Fc engineering to disfavour preferential Fc crystallisation. The IdeS protease displays extensive Fc recognition and encases the antibody hinge. Conversely, the glycan hydrolase domain in EndoS traps the Fc glycan in a "flipped-out" conformation, while additional recognition of the Fc peptide is driven by the so-called carbohydrate binding module. In this work, we reveal the molecular basis of antibody recognition by bacterial enzymes, providing a template for the development of next-generation enzymes.

The bacterium *Streptococcus pyogenes* has evolved a diverse range of mechanisms for evading the human adaptive immune system[1]. Infection with *S. pyogenes* can be mild, causing for example throat infections, but at the other extreme can cause terminal necrotising fasciitis[2]. Two enzymes secreted by this bacterium, IdeS[3] and EndoS[4], directly target and cleave IgG antibodies, and thereby impede cellular responses through immune recruitment mediated by the antibody Fc domain. The specificity of these enzymes for IgG has led to the development of a wide range of clinical and biotechnology applications[5] and has warranted extensive studies of their enzymology.

Of the two immune evasion factors, IdeS is most advanced in clinical development[6–8]. *S. pyogenes* expresses two variants of this enzyme (often distinguished by naming the first and second variants IdeS/Mac-1 and Mac-2, respectively), which display less than 50 % sequence identity within the middle third of the protein[9], but nonetheless exhibit largely indistinguishable endopeptidase activity[10]. The enzyme targets IgG by cleaving within the lower hinge region, yielding F(ab')$_2$ and Fc fragments[3,11,12], an activity which has enabled its development

(specifically, the Mac-1 enzyme variant) as a pre-treatment for transplantation in hypersensitised individuals with chronic kidney disease (Imlifidase, brand name Idefirix®)[6–8]. Along with EndoS, it has further potential use in the deactivation of pathogenic antibodies in autoimmune disorders[13–19], deactivation of neutralising antibodies for in vivo gene therapy[20], and for the potentiation of therapeutic antibodies by deactivation of competing serum IgG[21,22]. Imlifidase has also been used in combination with EndoS for inactivation of donor-specific antibodies in murine allogeneic bone marrow transplantation[23].

The endoglycosidase EndoS has additional biotechnological applications in engineering antibody glycosylation:[24] it hydrolyses the β−1,4 linkage between the first two N-acetylglucosamine (GlcNAc) residues within biantennary complex-type N-linked glycans on IgG Fc, thereby removing the majority of the glycan[4]. The related enzyme EndoS2 from serotype M49 of *S. pyogenes* also targets IgG[25] but exhibits broader glycan specificity[26]. Variants of both enzymes have been utilised in transglycosylation of various glycoforms to intact IgG to enable precise antibody glycan remodelling[24,27–29].

[1]School of Biological Sciences, University of Southampton, Southampton SO17 1BJ, UK. ✉e-mail: Ivo.Tews@soton.ac.uk; Max.Crispin@soton.ac.uk

It is still unclear, however, how exactly these enzymes specifically target and degrade IgG. Full cleavage of an antibody by IdeS occurs in two distinct steps, in which the second chain is cleaved more slowly;[11,12] this observation, along with the finding that IdeS exhibits low activity towards synthetic hinge peptides[30], suggests a more extensive recognition interface with the target IgG. Similarly, multiple domains within EndoS contribute to substrate recognition and catalysis[31–33], but the molecular details of substrate recognition remain undefined.

Here, we illustrate the molecular basis behind the unique substrate specificity of these enzymes using X-ray crystallography. We show that mutation of IgG Fc residue E382, which consistently forms salt bridge interactions in Fc crystal structures, can be used as a strategy to discourage Fc self-crystallisation and thus promote crystallisation of protein complexes. We present crystal structures of IdeS/IgG1 and EndoS/IgG1 complexes, to a resolution of 2.34 Å and 3.45 Å, respectively, and map the extensive interfaces that are formed in these complexes. Understanding substrate interaction and recognition of these enzymes facilitates their further clinical development and their application as highly specific biotechnological tools.

## Results and discussion
### Analysis of Fc crystal structures for Fc engineering
The co-crystallisation of IgG Fc with enzymes is notoriously difficult, due to the inherent ability of the Fc fragment to crystallise on its own. We therefore sought to identify favourable contacts present in typical Fc crystals, in order to devise a strategy to counteract its selective self-crystallisation.

We have observed, from looking at structures currently present in the PDB, that human IgG Fc commonly crystallises in the $P2_12_12_1$ space group (60.2 % of 108 apo IgG Fc structures, as of August 2022). We studied the crystal lattice contacts present in a typical, wild-type Fc structure (PDB 3AVE[34]), in order to identify amino acid residues which are important in this favourable packing arrangement. As calculated in PDBePISA[35], model 3AVE forms thirteen salt bridges and fifteen hydrogen bonds with neighbouring molecules within its crystal lattice (Fig. 1b). In addition, contacts are largely conserved across both Fc chains, resulting in a tight packing arrangement (Fig. 1b, c). We identified residue E382, which forms salt bridges with R255 in a neighbouring Fc molecule (and vice versa), in both Fc chains (Fig. 1b). We hypothesised that replacement of this residue would hinder the self-association of the Fc into this preferred crystal lattice, and therefore designed three IgG1 Fc variants: E382R, E382S and E382A, which we collectively term as "Fx" variants.

In order to compare the crystallisation abilities of our "Fx" variants versus a wild-type IgG1 Fc, we set up identical crystallisation experiments in JCSG-plus™ and Morpheus screens (both from Molecular Dimensions) for each Fc at 10 mg/mL. Crystals were left to grow at 21 °C and, after eight days, the number of conditions in each screen producing crystals were counted. Of the three Fx variants, we illustrate results here for the E382S variant. Wild-type IgG1 Fc produced a total of 21 crystal "hits" (14 in JCSG-plus™ and 7 in Morpheus). In contrast, the E382S variant produced 9 hits in total, all of which were in JCSG-plus™ (Fig. 1a); this variant therefore displayed a ~57 % reduction in crystallisation compared to the wild-type Fc. The E382A and E382R variants similarly produced no hits in the Morpheus screen, and yielded 8 and 6 hits in JCSG-plus™, respectively.

Crystals of the Fc E382S variant were found to have grown in an atypical space group $P3_221$. The structure was determined by molecular replacement using 3AVE as a search model and refined to a resolution of 3.04 Å (Supplementary Table 1, Supplementary Fig. 1). As of August 2022, this space group has not previously been reported for a human apo IgG Fc structure; we did, however, find examples of IgG Fc crystallised in complex with a small peptide (for example, PDB 5DVK[36]) where the reported space group was $P3_221$. Interestingly, this peptide binds at the Cγ2-Cγ3 interface within the Fc, around the same area as E382.

Analysis of the crystal contacts revealed that this variant makes fewer interactions with symmetry-related molecules in the crystal (four salt bridges and sixteen hydrogen bonds; Fig. 1b), which are asymmetrical across the two Fc chains, resulting in altered crystal packing (Fig. 1c). Furthermore, as calculated within the ccp4i2 interface[37], the E382S variant had a higher solvent content and Matthews coefficient compared to the wild-type Fc (Fig. 1b), indicating that the molecules are less tightly packed in this crystal form. We conclude that crystallisation of this Fc variant has been rendered less favourable; this indicates that other substitutions at E382, as in the E382R and E382A variants, would have similarly altered crystallisation. In general, we envisage that any mutation impacting lattice formation could be similarly employed. We subsequently used these E382 variants for screening of enzyme-Fc complexes, and believe that they would similarly be ideal for crystallisation of other Fc complexes, such as those of IgG with Fc γ-receptors.

### The IdeS-IgG1 Fc complex
IdeS from *Streptococcus pyogenes* (strain MGAS15252), containing a C94A mutation to abolish catalytic activity, was combined in a 1:1 molar ratio with our panel of IgG1 Fc variants, and the resulting complexes were purified by size exclusion chromatography (Supplementary Fig. 2). We obtained crystals of IdeS in complex with the Fc[E382A] variant, which crystallised in space group $C121$ (Supplementary Table 2). The structure was determined by molecular replacement with 1Y08 and 3AVE search models. The data for this crystal is twinned, with a refined twin fraction of 0.493 for operator $-h,-k,l$, as determined using twin refinement in Refmac5[38]. Such a high twin fraction means that significant model bias is to be expected, and care must be taken in model analysis (see Supplementary Table 2 and Supplementary Fig. 4).

Electron density resolves amino acids 43-339 in IdeS, as well as 229-445 and 230-444 for chains A and B in IgG1 Fc, respectively. We additionally observe density for seven/eight monosaccharide residues at the N-linked glycosylation site (at N297) on Fc chains A and B, comprising a fucosylated biantennary glycan with a single β–1,2-linked GlcNAc on the mannose 6-arm (chain A) and the equivalent glycan with terminal β–1,2-linked GlcNAc on both arms (chain B). The final structure was refined to 2.34 Å (Supplementary Table 2, Supplementary Fig. 3) and is depicted in Fig. 2.

The crystal structure shows asymmetric binding of IdeS across the Cγ2 domains of the Fc and its lower hinge region (Fig. 2a). We envisage that the upper hinge region of IgG and its Fab regions do not contribute significantly to complex formation, as indicated with the lack of electron density for the hinge above residues 229/230 (in Fc chains A and B, respectively), and reported cleavage of both full-length IgG and its Fc fragment by IdeS[30]. However, we cannot formally exclude the possibility that there is some interaction of IdeS with the IgG Fab regions.

The 1:1 stoichiometry observed in the crystal structure is consistent with previous kinetic analyses[39] showing that IdeS functions predominantly in a monomeric form. The enzyme appears to clamp down over the lower hinge region of one Fc chain (Fig. 2a), creating a cavity in which the catalytic residues are brought into close proximity with the cleavage site. Binding of the enzyme to the Fc appears to displace the two Cγ2 domains slightly, as shown by superposition with a structure of wild-type IgG1 Fc (PDB 3AVE) (Supplementary Fig. 4b). Residues within the Cγ2 domain in chain A have higher B factors compared to the rest of the complex (Supplementary Fig. 4a), which could indicate that binding of IdeS pulls this domain away slightly from the rest of the antibody.

### Role of prosegment binding loop in IdeS-Fc complex
IdeS crystallised in complex with IgG1 Fc here is the Mac-2 variant, and thus deviates in sequence from the three published apo structures of IdeS (all of which are the Mac-1 variant; Supplementary Fig. 5a). Despite this, a structural alignment shows very few deviations (Fig. 2b). Complexed IdeS contains ten α-helices and twelve β-strands, as calculated

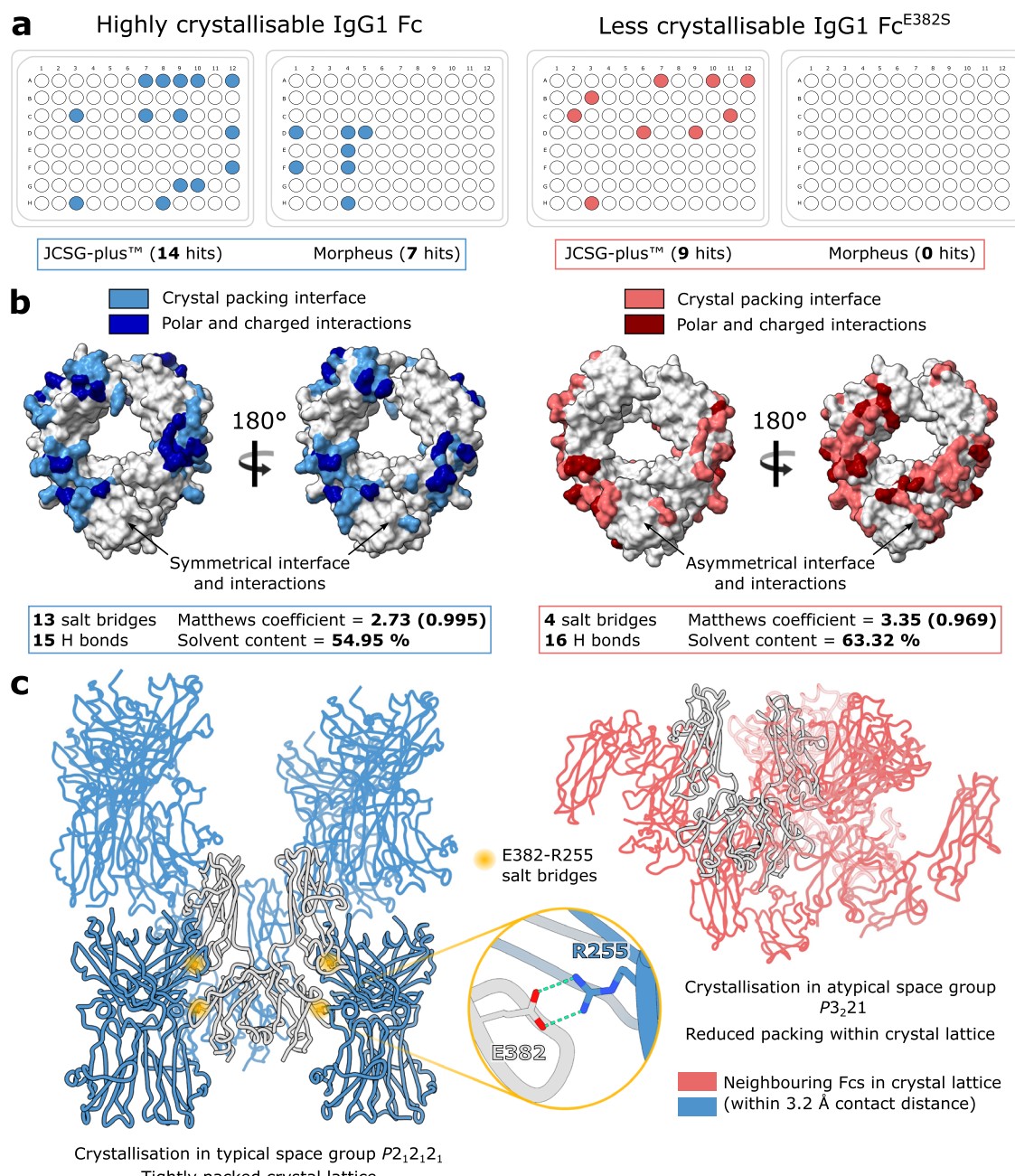

**Fig. 1 | Observed crystal packing in wild-type and "less crystallisable" IgG1 Fc fragments. a** Crystallisation of wild-type IgG1 Fc and IgG1 Fc[E382S] variant in JCSG-plus™ and Morpheus screens, at 10 mg/mL and 21 °C. Crystal "hits" are indicated with a coloured circle. **b** Analysis of crystal packing interface and interactions present in a typical, wild-type IgG1 Fc crystal structure (PDB ID 3AVE) and IgG1 Fc[E382S] variant, as calculated by PDBePISA[35]. **c** Crystal packing resulting from crystallisation in typical space group $P2_12_12_1$ and atypical space group $P3_221$, for the wild-type IgG1 Fc and IgG1 Fc[E382S] variant, respectively. E382-R255 salt bridges between symmetry-related Fcs in the $P2_12_12_1$ crystal lattice are highlighted with a yellow circle. Neighbouring Fcs in the crystal lattice contacting the origin Fc within a 3.2 Å contact distance are shown. **a–c** Analysis relating to the wild-type IgG1 Fc and IgG1 Fc[E382S] variant is depicted in blue and red, respectively.

by DSSP[40,41] (Supplementary Fig. 4c); we note that the loop located between β-strands seven and eight is modelled in distinct conformations for each of the apo structures[42] and is not included within 1Y08[43] (Fig. 2b), signifying its inherent flexibility in the apo form. This loop is equivalent to the "prosegment binding loop" present in other papain superfamily cysteine proteases; in these enzymes, which are synthesised as inactive zymogens, this loop packs against the prosegment as a mode of inhibition[44–46]. In complexed IdeS, the loop curls upwards to accommodate the Fc hinge within the active site cavity (Fig. 2a).

Alanine substitution mutations within this loop were previously found to have little effect on neither IdeS binding to IgG, nor its catalytic activity[42]. Our structure shows, however, that the majority of interactions present here involve the IdeS backbone, whose conformation won't be significantly altered by alanine mutations. The inability of IdeS to cleave IgG hinge-mimicking peptides[30] also indicates an occlusion of the active site in the absence of substrate, especially given the strong potential of hydrogen bonding and hydrophobic interactions observed at the Fc hinge (discussed in the following section). It is possible that there may be a conformational change in the active site upon binding; however, superposition of wild-type IdeS (PDB 2AU1) with the complexed enzyme shows the catalytic triad residues in very similar conformations, although we cannot rule this out given the sequence diversity present (Supplementary Fig. 5). Moreover, IdeS[C94A] has been shown to retain antibody binding and

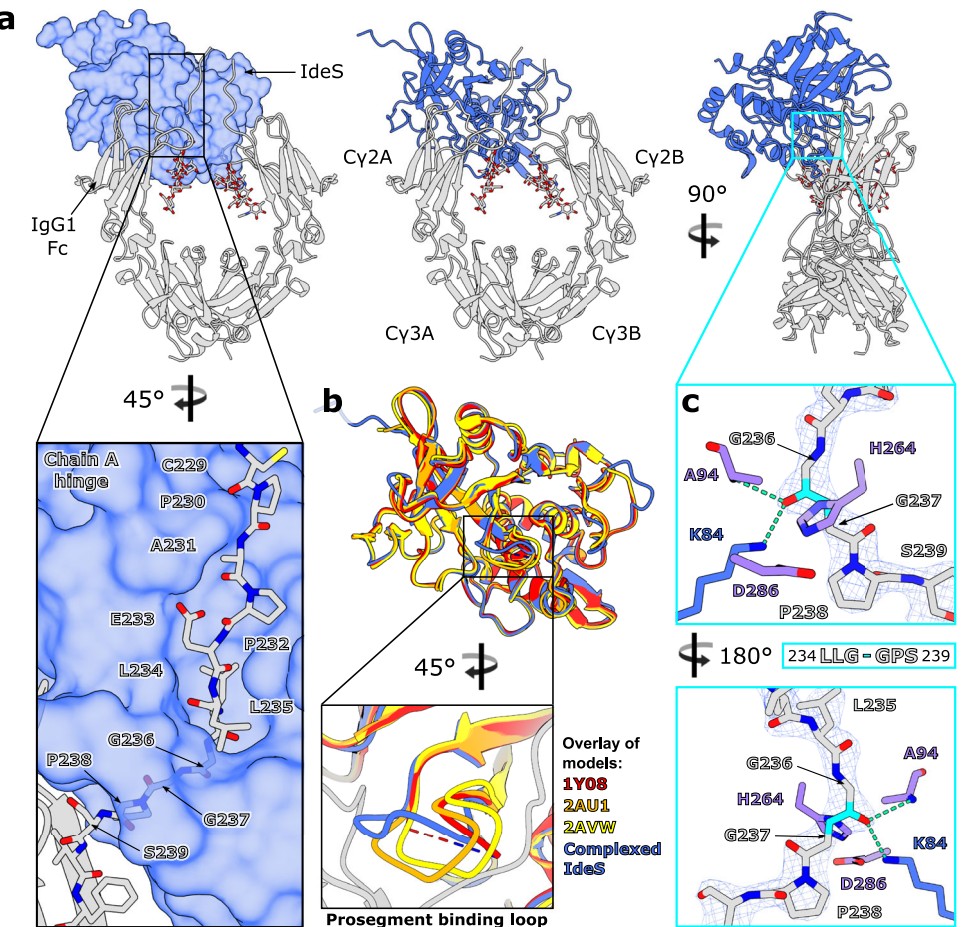

**Fig. 2 | Crystal structure of IgG1 Fc^E382A-IdeS^C94A complex. a** Overall structure of complex, with IdeS^C94A shown as a surface and IgG1 Fc^E382A shown as a cartoon. N-linked glycans and the IgG1 Fc^E382A hinge peptide in the focused panel are shown as sticks and coloured with oxygen, nitrogen and sulphur atoms in red, blue and yellow, respectively. **b** Superposition of complexed IdeS^C94A with three published apo structures of IdeS (PDB IDs 1Y08, 2AU1 and 2AVW, coloured in red, orange and yellow, respectively) and focused view of the prosegment binding loop. **c** Binding of IgG1 Fc^E382A hinge peptide within the IdeS^C94A active site. Fc^E382A peptide and IdeS^C94A active site residues are depicted as sticks and coloured by heteroatom; catalytic triad residues are coloured purple. The scissile peptide bond is coloured in cyan; hydrogen bonds are depicted as green dashes. The final $2F_{obs}$-$F_{calc}$ electron density map corresponding to the Fc^E382A peptide is shown (weighted at 1.5 σ). **a–c** IdeS^C94A is coloured blue; IgG1 Fc^E382A is coloured in silver.

inhibit IgG-mediated phagocytosis at levels comparable to the wild-type enzyme[9], suggesting that the inactive enzyme retains antibody binding. We therefore conclude that the likely role of this loop is mediation of substrate access to the active site.

### Interaction of IgG1 Fc hinge at IdeS active site

We observe clear density for the IgG1 Fc hinge region bound within the IdeS active site cavity, in both the final electron density map (Fig. 2c) and a polder map as calculated in *PHENIX*[47,48] (Supplementary Fig. 4d). The carbonyl oxygen of G236 is hydrogen bonded to the amide nitrogen of the catalytic cysteine (mutated to alanine here) and the side chain of K84, which collectively form the oxyanion hole, as predicted[42,43]. Binding of the hinge distorts the Fc peptide backbone at G236 in order to promote scissile bond cleavage (Fig. 2c); this residue is thus identified in Molprobity[49] as a Ramachandran outlier. Super-position of wild-type IdeS (PDB 2AU1) with the complexed enzyme gives an indication for placement of the catalytic cysteine side chain (Supplementary Fig. 5b): in this conformation, the cysteine sulphur is ideally poised for nucleophilic attack on the carbonyl carbon within the scissile peptide bond.

### Extended exosite binding of IdeS to the Fc Cγ2 domains

It has long been suspected that IdeS must recognise its sole substrate IgG with exosite binding[30,42,43]. Our structure now reveals that IdeS

binds across both chains of the Fc region (Fig. 3a). Unsurprisingly, the most extensive interface is formed with the Fc chain being cleaved (annotated as chain A in our structure) (Fig. 3b), with an interface area of 1392 Å² and a solvation free energy gain upon interface formation of −16.2 kcal/mol, as calculated by PDBePISA[35]. The interface extends across the entire hinge region (C229-S239; Fig. 3b), with hydrogen bonds formed with the backbone at A231, L234, G236 and G237 and the E233 side chain, and favourable hydrophobic interactions predicted here (inferred by positive solvation energies of hinge residues). Within the Fc Cγ2 domain, IdeS interacts with residues in proximity of the BC loop, which aids in stabilising an "open" conformation of the prosegment binding loop (as discussed above), and additionally the FG loop (Fig. 3b).

A secondary interface is formed across the second Fc chain (annotated as chain B in our structure; Fig. 3c), with an interface area of 804.7 Å² and a solvation free energy gain of −7.6 kcal/mol. A smaller proportion of the Fc hinge contributes (A231-G237), but PDBePISA predicts favourable hydrophobic interactions here, albeit not to the same extent as chain A. Subsequent recognition of this Fc chain is driven by interactions with the BC loop, and, in contrast to chain A, the C'E loop containing the N-linked glycan (Fig. 3c). PDBePISA additionally predicts a small number of interactions between the enzyme and the Fc N-linked glycans; the lack of electron density for any monosaccharides past β−1,2-linked GlcNAc suggests that any further glycan

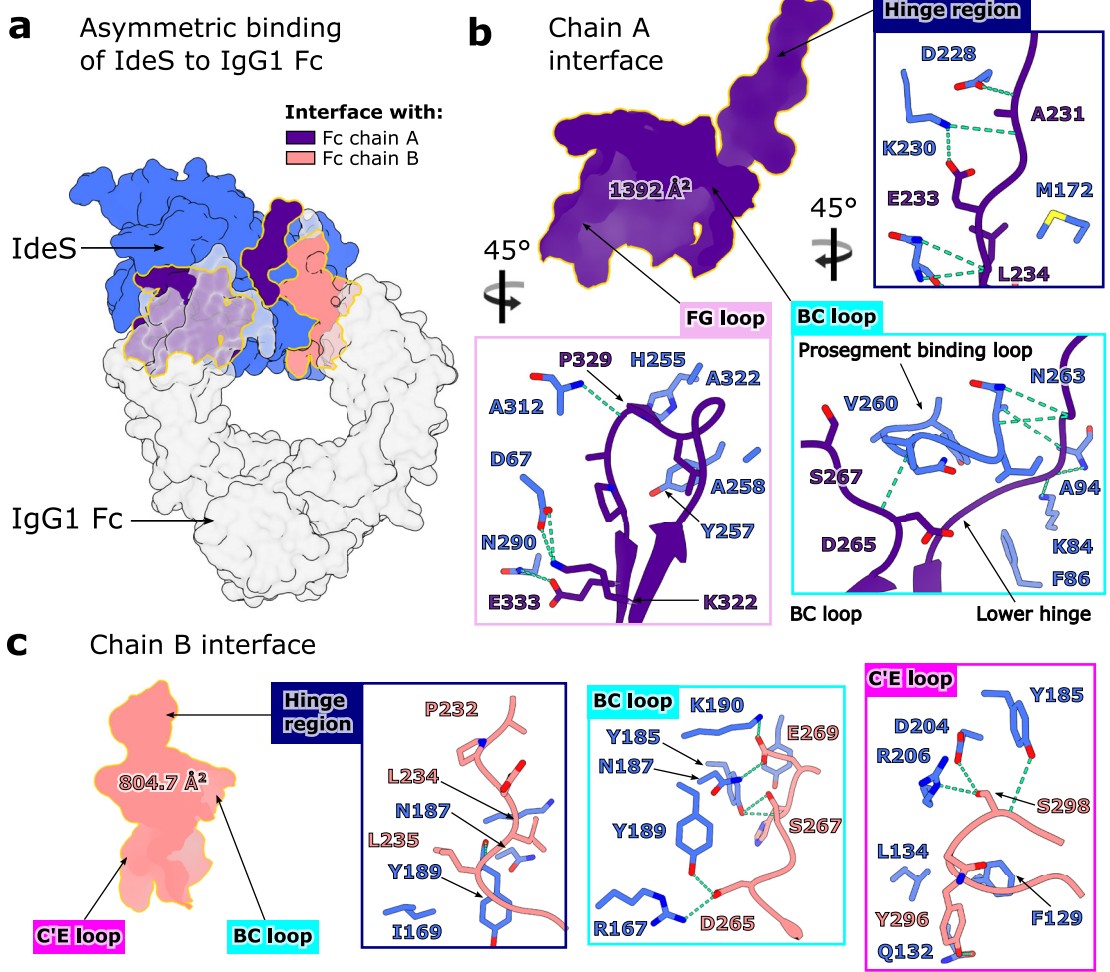

**Fig. 3 | Asymmetric binding interface of IdeS$^{C94A}$-IgG1 Fc$^{E382A}$ complex. a** Overall view of complex depicted as a surface, with IdeS$^{C94A}$ coloured blue and IgG1 Fc$^{E382A}$ coloured silver. Interfaces of IdeS$^{C94A}$ with chains A and B of the Fc are coloured indigo and coral, respectively. Glycans within the Fc have been omitted for clarity. **b** Interface between IdeS$^{C94A}$ and IgG1 Fc$^{E382A}$ chain A, involving the Fc hinge region, BC loop and FG loop. **c** Interface between IdeS$^{C94A}$ and IgG1 Fc$^{E382A}$ chain B, involving the hinge region, BC loop and C′E loop. **b**, **c** Residues involved in binding are depicted as sticks and coloured by heteroatom (oxygen in red and nitrogen in blue), with hydrogen bonds depicted as green dashes.

processing doesn't affect complex formation, and that IdeS can accommodate IgG with heterogenous glycosylation.

Although IdeS interacts with both chains in the Fc hinge simultaneously, following cleavage of the first chain, the complex would need to dissociate before the second cleavage could occur. This observation is also evidenced by detection of single-cleaved Fc in enzymatic assays and in clinical studies[11,39,50,51]. We suspect that the binding interface is altered for single-cleaved Fc and that this explains its slower rate of cleavage[11,12,39]. It is also interesting to note that, aside from the hinge region, IdeS binds Fc regions implicit in Fcγ-receptor binding, an observation also inferred by its ability to counteract Fc-mediated effector functions by competitive binding inhibition[9]. Moreover, we observe that IdeS residues interacting with the Fc are largely conserved across the two IdeS isoforms, and any substitutions are mostly to similar amino acids, which aids in explaining their near identical activity[10].

**The EndoS-IgG1 Fc complex**
To date, there are several known structures of endoglycosidases in complex with their glycan substrates[26,33,52–54]. Here, we present the structure of truncated EndoS (residues 98-995, as described previously[31]) in complex with its IgG1 Fc substrate (E382R variant). A catalytically inactive version of EndoS was generated by the inclusion

of D233A/E235L substitutions, as described previously[33]. The complex, containing two copies of EndoS and one IgG1 Fc molecule, crystallised in space group $P2_12_12_1$ and was refined to a resolution of 3.45 Å (Supplementary Table 3, Supplementary Fig. 6). The second copy of EndoS (annotated as chain D) is much more poorly resolved in the electron density compared to the rest of the structure (discussed below), and thus we have used the more ordered copy of EndoS (annotated as chain C) for structure depiction and analysis. In addition, the density resolves only the N-linked glycan on chain A of the Fc (not its counterpart on chain B), which is bound within the more ordered copy of EndoS. We therefore elucidate the mode of glycan binding and IgG recognition by inspecting the interaction of EndoS with this Fc chain. The final structure is depicted in Fig. 4.

Our structure of EndoS shows the same "V" shape as observed in its previously solved structures[31,33]. Following the previously-described domain classification[31], the structure comprises, from the N- to the C-terminus: a proline-rich loop (residues 98-112), a glycosidase domain (residues 113-445), a leucine-rich repeat domain (residues 446-631), a hybrid Ig domain (residues 632-764), a carbohydrate-binding module (CBM; residues 765-923) and a C-terminal three-helix bundle domain (C-3HB; residues 924-995) (Fig. 4a). One Cγ2 domain in IgG1 Fc (annotated as chain A in our structure) binds across the termini of the "V", in-between the glycosidase domain and CBM, with the rest of the

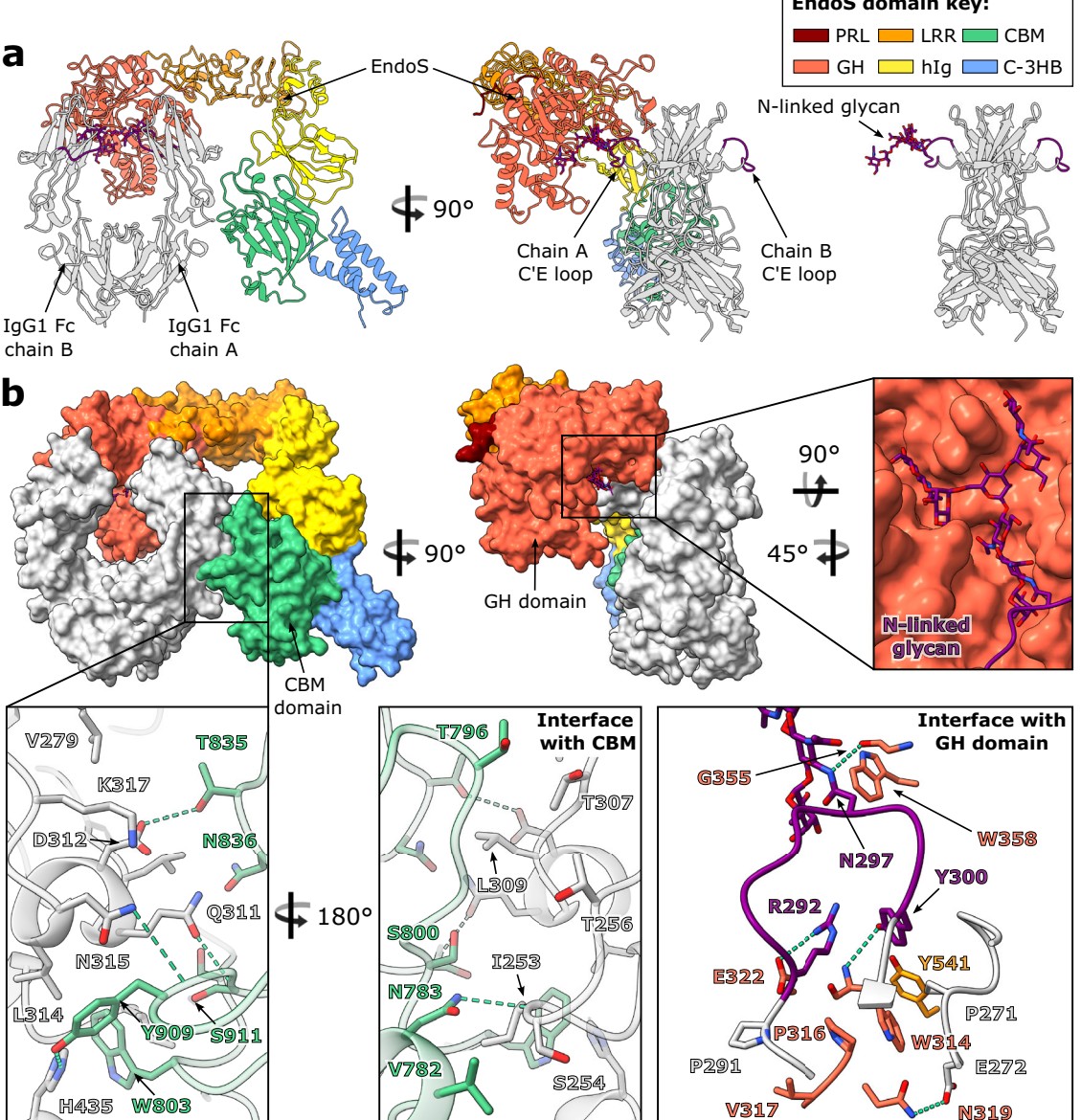

**Fig. 4 | Crystal structure of EndoS$^{D233A/E235L}$-IgG1 Fc$^{E382R}$ complex. a** Overall structure of complex depicted as a cartoon. IgG1 Fc$^{E382R}$ is coloured silver, with its C' E loops coloured purple; the N-linked glycan is shown as sticks and coloured by heteroatom (oxygen in red and nitrogen in blue). EndoS domains are coloured as follows: proline-rich loop (PRL), maroon; glycosidase domain (GH), red; leucine-rich repeat domain (LRR), orange; hybrid Ig domain (hIg), yellow; carbohydrate-binding module (CBM), green; C-terminal 3-helix bundle (C-3HB), blue. **b** EndoS$^{D233A/E235L}$-IgG1 Fc$^{E382R}$ complex depicted as a surface, highlighting binding to IgG1 Fc$^{E382R}$ by the CBM and GH domains. Residues involved in binding are depicted as sticks and coloured by heteroatom. Hydrogen bonds are depicted as green dashes.

antibody remaining exposed to the surrounding solvent. Previous work investigating the ability of EndoS to cleave the N-linked glycans from various Fc fragments in comparison to full-length IgG showed EndoS was able to cleave the majority of glycans in all instances, indicating that the IgG Fab regions are unimportant in complex formation[32]. However, as with the IdeS-IgG1 Fc structure, we are unable to exclude the possibility that EndoS interacts with the IgG Fab regions from this crystal structure alone.

The N-linked glycan on this chain is "flipped-out" from its usually-observed position between the two IgG Fc Cγ2 domains[34] and is bound within the previously-identified glycosidase domain cavity[33] (Fig. 4b). A structural overlay with full-length EndoS in complex with its G2 oligosaccharide substrate (PDB 6EN3[33]) shows that the overall morphology and domain organisation of EndoS is approximately maintained (Supplementary Fig. 7a),

apart from a slight shift of the CBM and C-3HB, likely due to a pinching of the CBM around the Fc as it binds.

## Role of CBM in governing specificity of EndoS for IgG

Our structure of the EndoS-Fc complex reveals how one Cγ2 domain of the Fc binds across the glycosidase domain and CBM (Fig. 4). As calculated by PDBePISA[35], the interface between chain A of the Fc and EndoS comprises an area of 1323.5 Å$^2$ and yields a solvation free energy gain of −9.1 kcal/mol. The glycosidase domain of EndoS is observed forming contacts with the glycan-containing C'E loop, while the CBM forms additional interactions at the Fc Cγ2-Cγ3 interface (Fig. 4b). We note that residue W803 within the CBM, whose substitution to an alanine has previously been shown to abolish hydrolytic activity against all human IgG subclasses[31], appears to act as a hydrophobic "plug"; it binds within a cavity at the Cγ2-Cγ3 interface containing Fc

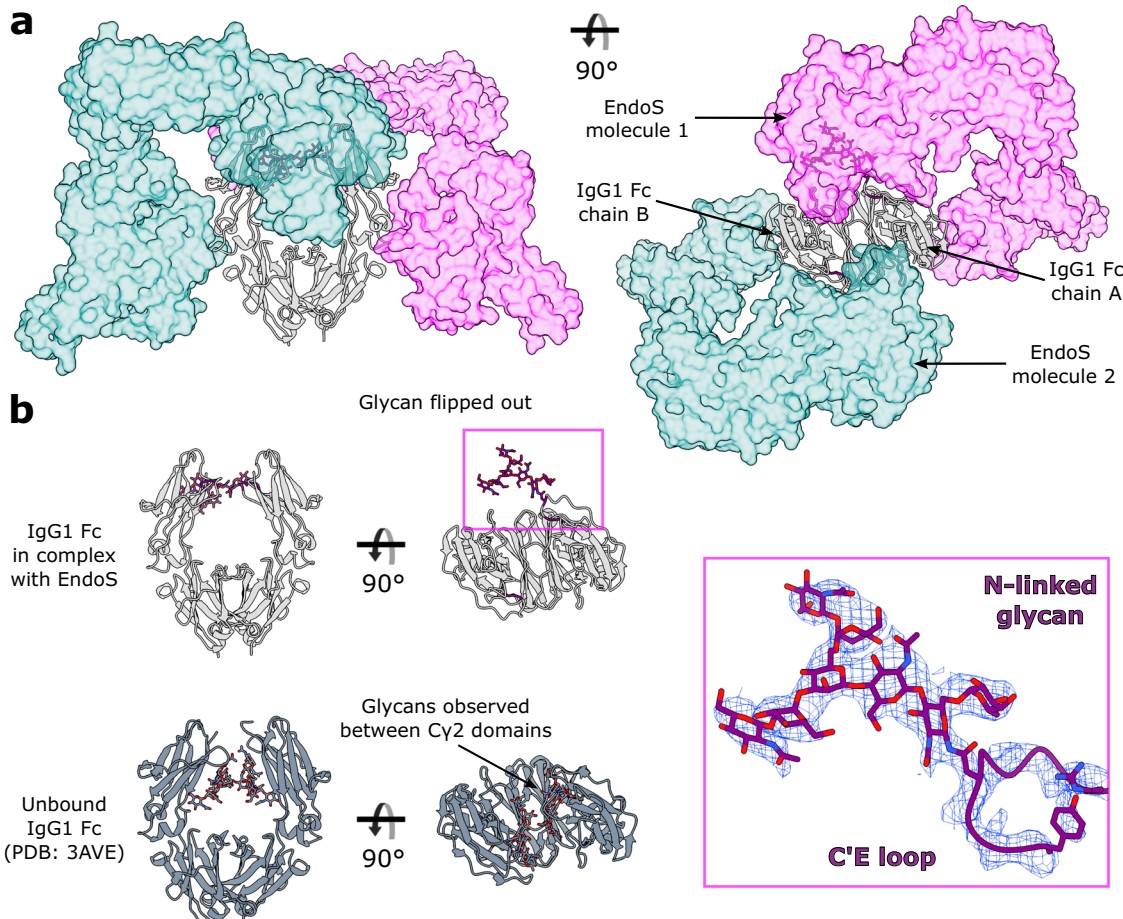

**Fig. 5 | Stoichiometry of the EndoS$^{D233A/E235L}$-IgG1 Fc$^{E382R}$ complex. a** One EndoS$^{D233A/E235L}$ molecule interacts with one chain within IgG1 Fc$^{E382R}$, resulting in an overall 2:2 binding stoichiometry. The two EndoS molecules are coloured teal and magenta and depicted as a surface at 50% transparency, while the Fc is coloured silver and depicted as a cartoon. C'E loops within the Fc are coloured purple; the N-linked glycan is depicted as sticks and coloured by heteroatom (oxygen in red and nitrogen in blue). **b** Comparison of N-linked glycan positions observed in IgG1 Fc$^{E382R}$ bound to EndoS$^{D233A/E235L}$, and a wild-type IgG1 Fc structure (PDB ID 3AVE, coloured in dark grey). N-linked glycan is observed in a "flipped-out" structure in the complexed Fc$^{E382R}$, while N-linked glycans in typical Fc structures are observed between the Fc Cγ2 domains. Electron density from the final $2F_{obs}$-$F_{calc}$ map, corresponding to the glycan and C'E loop, is shown (weighted at 1.1 σ).

residues I253, H310, L314 and H435 (Fig. 4b), and has the highest solvation energy (of 2.06 kcal/M) of all EndoS residues calculated by PDBePISA, indicating that strong hydrophobic interactions are present here. A small number of contacts are also predicted between EndoS and the second Fc Cγ2 domain, although these are unlikely to be necessary for complex formation, given that EndoS can cleave the Fc Cγ2 lacking the hinge region (likely monomeric)[32].

The complex structure presented here corroborates previous findings that both the glycosidase domain and the CBM are important for IgG Fc binding[31] and glycan hydrolysis[32], and that EndoS can cleave the Cγ2 homodimer fragment of IgG Fc[32]. The related enzyme EndoS2 likely binds IgG in a similar manner;[25] hydrogen-deuterium exchange mass spectrometry on this complex has similarly indicated strong binding of IgG to the glycosidase domain and the CBM[26]. While mutation of residues within the glycan binding site of both enzymes completely abolishes their hydrolytic activity[26,33], EndoS lacking the CBM can still hydrolyse IgG, albeit at greatly reduced capacity[31,32]. Therefore, the CBM appears to drive additional specificity of EndoS for the Fc peptide surface.

Interestingly, although the CBM was assigned based on homology to a legitimate carbohydrate-binding domain[31] and previous work has indicated that it has the capacity to bind galactose (albeit with low affinity)[32], here the CBM is not observed to bind the Fc N-linked glycan. Furthermore, the N- and C-terminal 3 helix bundles, which are homologous to IgG-binding protein A from *Staphylococcus aureus*[33,55], are not interacting with the substrate polypeptide within this complex. A structural overlay of complexed EndoS with full-length EndoS (PDB 6EN3) indicates that the N-terminal bundle would not contact the Fc (Supplementary Fig. 7a), thus its contribution to EndoS-IgG binding and glycan hydrolysis is likely solely due to stabilisation of the glycosidase domain, as suggested previously[33]. Indeed, existence of the crystal structure is evidence in itself that EndoS forms a stable complex with IgG in the absence of the N-terminal bundle.

**Stoichiometry of the EndoS-IgG Fc complex**
Within the crystal, each of the two IgG Fc chains binds a distinct EndoS molecule, resulting in a complex with 2:2 stoichiometry (Fig. 5a). The first EndoS molecule (chain C) is binding chain A of the Fc, and we observe clear electron density for the N297 glycan binding within the EndoS glycosidase domain cavity previously identified[33] (Fig. 5b). The polder map for this carbohydrate group, calculated in *PHENIX*[47,48], supports the presence of an uncleaved N-linked glycan in the substrate binding pocket of EndoS (Supplementary Fig. 8). This observation of an Fc glycan in this "flipped-out" conformation is in strong contrast to typical crystal structures of IgG Fc, whose N-linked glycans are interspersed between the Cγ2 domains[34] (Fig. 5b).

Chain B of the Fc appears to be binding a second EndoS molecule (chain D; Fig. 5a); however, this EndoS molecule is poorly resolved in

the electron density, with higher $B$ factors and a greater proportion of residues identified as RSRZ outliers (Supplementary Fig. 9). Moreover, the Fc N297 glycan in chain B is not visible in the electron density, although the second EndoS molecule appears to bind this Fc chain in the same manner as its more ordered counterpart (Fig. 5a). Electron density for the Fc C′E loop in chain B (albeit less clear than chain A) also indicates that the glycan is in close proximity to the glycosidase domain in the second EndoS molecule.

It is fascinating to observe the glycan trapped in this "flipped-out" conformation, and this substantiates several recent studies documenting the existence of IgG Fc glycan conformational heterogeneity[56–60]. Superposition of this complexed IgG with a wild-type Fc (PDB 3AVE) illustrates that movement of the glycan into this position is governed by movement of the C′E loop only (Supplementary Fig. 7b), although it is possible that the lower resolution of the data is masking small chain shifts. The observation of a "flipped-out" glycan conformation may also provide a structural explanation for the ability of cellular glycosidases and glycosyltransferases to process this otherwise sterically-restricted substrate. Moreover, it appears that the capture of Fc N-linked glycans in this state allows space for two enzymes to bind simultaneously; however, there is no evidence to suggest that this 2:2 assembly is required for activity, especially given previous work showing that EndoS is largely monomeric in solution[31,33]. Although EndoS crystallised here is lacking the N-terminal 3-helix bundle, a structural superposition with full-length EndoS (Supplementary Fig. 7a) suggests 2:2 binding would be able to occur in its presence.

The crystal structures presented here provide a structural rationale for the unique properties of these two enzymes, particularly their exquisite substrate specificity towards human IgG. Understanding the molecular basis of this activity is critical for expanding their clinical and biotechnological use. For example, the deactivation of serum IgG using both IdeS and EndoS can strengthen the potency of therapeutic antibodies;[21,22] this strategy could be applied to potentiate any therapeutic antibody, in theory, if the antibody were designed to be resistant to cleavage by these enzymes, a venture which can be aided greatly with structural information. This will also be invaluable in the synthesis of immunologically-distinct enzyme variants which retain identical activity, for their long-term therapeutic use. While EndoS variants have already been designed to expand the ability to engineer antibody glycosylation[27–29], the structural information presented here will allow this to be extended further. We also envisage that this structural information will help in the development of anti-streptococcal biologics resistant to enzyme-mediated degradation. To conclude, this work will assist in the continued development of IdeS and EndoS as enzymatic tools with wide clinical and biotechnological applications.

## Methods

### Cloning, expression and purification of IdeS/EndoS
Gene fragments encoding IdeS$^{C94A}$ (amino acids 41-339, gene accession number AFC66043.1) and EndoS$^{D233A/E235L}$ (amino acids 98-995, as described previously[31,33], genome accession number AP012491.2) were synthesised to contain a C-terminal linker and His tag (sequence LEHHHHHH), and cloned into pET21a(+) vectors by NBS Biologicals. Constructs for truncated, inactive IdeS and EndoS used for IgG complex crystallography are shown in Supplementary Fig. 10. Constructs were expressed in *E. coli* BL21 (DE3)pLysS cells (Thermo Fisher). Cells were grown at 37 °C in Terrific Broth (Melford) in the presence of 100 μg/mL ampicillin and 34 μg/mL chloramphenicol, until an OD$_{600}$ of 0.8 was reached, when protein expression was induced by addition of 1 mM IPTG. Cells were left to shake overnight at 25 °C, 200 rpm (Innova 43 R incubator; New Brunswick Scientific). Cells were collected by centrifugation at 6220 × $g$ for 20 min, resuspended in PBS containing 2 μg/mL DNAse1 (Sigma) and a pinch of lysozyme (Sigma),

homogenised using a glass homogeniser and broken apart using a cell disruptor (Constant Cell Disruption Systems). The remaining sample was centrifuged first at 3100 × $g$ for 20 min, then again at 100,000 × $g$ for one hour, to remove remaining cell debris and cell membranes. The resulting supernatant was subsequently filtered through a 0.2 μm membrane. Proteins were purified from the supernatant using Ni affinity chromatography with a HisTrap HP column (Cytiva) followed by size exclusion chromatography with a Superdex 75 16/600 column (Cytiva), equilibrated in 10 mM HEPES, 150 mM NaCl, pH 8.0.

### Cloning, expression and purification of IgG1 Fcs
IgG1 Fcs were expressed from a pFUSE-hIgG1-Fc vector (encoding residues 221-447 of IgG1; plasmid purchased from InvivoGen). Wild-type IgG1 Fc was expressed exactly as encoded within this plasmid; mutations for the E382A/S/R constructs were introduced by site-directed mutagenesis (QuikChange II kit; Agilent), using mutagenic primers (Supplementary Table 4) synthesised by Eurofins Genomics. Sequences of resulting IgG1 Fc constructs are shown in Supplementary Fig. 11. Fcs were transiently expressed in FreeStyle293F cells (ThermoFisher), using FreeStyle™ MAX Reagent (ThermoFisher), as described in the manufacturer's protocol. Cells were left to incubate at 37 °C, 8% CO$_2$, shaking at 125 rpm (New Brunswick S41i incubator), and harvested after seven days by centrifugation at 3100 x $g$ for 30 minutes. Supernatants were filtered through a 0.2 μm membrane and antibodies purified by affinity purification with a HiTrap Protein A HP column (Cytiva), followed by size exclusion chromatography with a Superdex 200 16/600 column (Cytiva) in 10 mM HEPES, 150 mM NaCl (pH 8.0).

### Quantification of IgG1 Fc crystallisation
Wild-type IgG1 Fc and Fc E382A/S/R variants were expressed and purified as detailed above. Identical sitting drop vapour diffusion crystallisation trays were set up using an Oryx4 robot (Douglas Instruments), in JCSG-plus™ and Morpheus crystallisation screens (Molecular Dimensions), using Fcs concentrated to 10 mg/mL. Crystals were left to grow at 21 °C for eight days, after which the number of crystal "hits" were counted (a "hit" constitutes crystals observed growing in a particular condition).

### Protein crystallisation
Protein concentrations were determined with a DS-11+ Spectrophotometer (DeNovix), using molecular weight and extinction coefficients calculated by the ProtParam tool[61]. IdeS/EndoS were combined with IgG1 Fcs in a 1:1 molar ratio and applied to a Superdex 200 16/600 column (Cytiva) equilibrated in 10 mM HEPES, 150 mM NaCl (pH 8.0). Fractions corresponding to the main peak only were pooled for crystallisation. Purified complexes were exchanged into 50 mM HEPES, 150 mM KCl (pH 7.5) prior to crystallisation, using a Vivaspin 20 centrifugal concentrator (MWCO 30 kDa; Sigma). Sitting drop vapour diffusion crystallisation trays were set up using an Oryx4 robot (Douglas Instruments). Crystals of the IdeS$^{C94A}$-IgG1 Fc$^{E382A}$ complex were grown in 0.12 M monosaccharides mix, 0.1 M buffer system 3 (pH 8.5), 30 % v/v precipitant mix 1 (Morpheus crystallisation screen; Molecular Dimensions). Crystals of the EndoS$^{D233A/E235L}$-IgG1 Fc$^{E382R}$ complex were grown in 0.09 M halogens, 0.1 M buffer system 2 (pH 7.5), 37.5 % v/v precipitant mix 4 (Morpheus crystallisation screen; Molecular Dimensions). Crystals of IgG1 Fc$^{E382S}$ were grown in 0.2 M ammonium sulphate, 0.1 M Tris (pH 7.5), 25 % w/v PEG 8000. Crystals were cryoprotected in mother liquor with 20 % glycerol added and flash-frozen in liquid nitrogen prior to data collection.

### Data collection and structure determination
Data collection for the IdeS$^{C94A}$-IgG1 Fc$^{E382A}$ complex was carried out at the European Synchrotron Radiation Facility (Grenoble, France) on beamline ID30A-3, under a 100 K cryostream ($\lambda$ = 0.9677 Å). Data

collection for the IgG1 Fc$^{E382S}$ variant was carried out at the European Synchrotron Radiation Facility on beamline ID30A-3, also at 100 K ($\lambda = 0.968$ Å). Data collection for the EndoS$^{D233A/E235L}$-IgG1 Fc$^{E382R}$ complex was carried out at Diamond Light Source (Oxford, UK) on beamline I03, at 100 K ($\lambda = 0.9763$ Å). Data processing of diffraction images was carried out using DIALS[62] and XDS[63]. Structures were solved by molecular replacement with the program Molrep[64]. 3AVE was used as a search model to solve the IgG1 Fc$^{E382S}$ structure; IdeS$^{C94A}$-IgG1 Fc$^{E382A}$ and EndoS$^{D233A/E235L}$-IgG1 Fc$^{E382R}$ were solved using initial search models for the enzyme (PDB ID 1Y08 for IdeS; 6EN3 for EndoS), after which the resulting solution was used as a fixed model for a second round of molecular replacement, using 3AVE as the search model. Models were improved with successive rounds of model building and refinement, using Coot[65] and Refmac5[38], respectively, within the ccp4i2 suite[37]. Due to the presence of twinning in the IdeS$^{C94A}$-IgG1 Fc$^{E382A}$ data, this structure was refined with the option for twinning ticked in Refmac5 and converged to a twin fraction of 0.493 for operator *-h,-k,l*. All structures were refined using local non-crystallographic symmetry restraints. Electron density maps for the EndoS$^{D233A/E235L}$-IgG1 Fc$^{E382R}$ model were calculated using map sharpening in Refmac5. The PDB-REDO[66] server was used to generate restraints for the IgG1 Fc$^{E382S}$ model for use in refinement. MolProbity[49] and the PDB validation server[67] were used for model validation prior to deposition. Carbohydrates were modelled in Coot[68] and validated using Privateer[69]. Data collection and refinement statistics for IgG1 Fc$^{E382S}$, IdeS$^{C94A}$-IgG1 Fc$^{E382A}$ and EndoS$^{D233A/E235L}$-IgG1 Fc$^{E382R}$ models are presented in Supplementary Tables 1–3, respectively. Polder maps were calculated with the *phenix.polder* tool[48] within the *PHENIX* software suite[47] using default settings. Protein complex interfaces were analysed using PDBePISA[35]. UCSF ChimeraX[70] was used to prepare figures depicting protein structure.

### Reporting summary

Further information on research design is available in the Nature Portfolio Reporting Summary linked to this article.

## Data availability

The data that support this study are available from the corresponding author upon request. Structure factor files and atomic coordinates for each of the crystallographic models presented in this study have been deposited in the PDB, with accession codes 8A47, 8A48, and 8A49, for the IdeS$^{C94A}$-IgG1 Fc$^{E382A}$ complex, IgG1 Fc$^{E382S}$ and the EndoS$^{D233A/E235L}$-IgG1 Fc$^{E382R}$ complex, respectively. Previously-elucidated crystal structures analysed in this study are deposited in the PDB, with accession codes 3AVE, 1Y08, 2AU1, 2AVW and 6EN3, which correspond to wild-type IgG1 Fc, IdeS$^{C94S}$, wild-type IdeS, IdeS$^{C94A}$ and EndoS$^{D233A/E235L}$, respectively.

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

## Acknowledgements

We are grateful to the beamline scientists on I03 at Diamond Light Source (DLS) and ID30A-3 at the European Synchrotron Radiation Facility (ESRF), and to Chris Holes for his support with the macromolecular crystallisation facility at the University of Southampton. This work was supported by DLS, ESRF (MX2373) and the School of Biological Sciences, University of Southampton.

## Author contributions

M.C. conceived the project. I.T. and M.C. supervised the project. A.S.L.S., J.B., D.P.I. and I.T. performed experimental work. A.S.L.S., I.T. and M.C. performed computational work. A.S.L.S. prepared the original draft together with initial data presentation. I.T. and M.C. reviewed and edited the manuscript. All authors have read and approve the final manuscript.

## Competing interests

The authors declare no competing interests.
