## [Peer Review File · Nature Communications]

REVIEWER COMMENTS

Reviewer #1 (Remarks to the Author):

This study by Sudol et al, attempts to elucidate the structural mechanisms behind antibody glycan hydrolysis and proteolysis by the antibody-specific endoglycosidase EndoS and IgG protease IdeS from *Streptococcus pyogenes*. These two enzymes are quite unique among glycosidases and proteases since they in contrast to the majority of these very specifically hydrolyze glycan and the protein backbone, respectively, on IgG antibodies. These activities are of major biological and medical significance, since *S. pyogenes* is a major human pathogen, but maybe even more importantly since both enzymes are used/being developed as drugs conditions such as transplant and/or biotechnological for analysis and modification of antibodies in research and for antibody drug development.

In order to gain further insight into the structural basis for the extreme IgG specificity of both enzymes, this team of researchers attempted to solve the crystal structures of IdeS and EndoS in complex with their IgG Fc substrate. In order to achieve this, they used a very clever approach to first control Fc's propensity to crystalize on its own. By looking at known Fc structures and mutagenesis, IgG1 Fc variants with reduced tendency to self-associate were generated. These were then co-crystallized with enzymatically inactive forms of IdeS and EndoS.

The overall structure of IdeS-IgG1 Fc shows asymmetric binding of IdeS across the C γ 2 domains predominantly in a monomeric form. Furthermore, a loop between beta strands seven and eight that is flexible in IdeS alone, curls upwards in the co-crystal to accommodate the IgG hinge within the active site cavity. The crystal structure also confirms interaction between the Fc hinge and the active site in IdeS. Previously, exosite binding between IgG and IdeS providing specificity has been hypothesized. The crystal revealed binding of IdeS across both chains of Fc which could explain the earlier observed slower hydrolysis of monomeric Fc. Interestingly, it also shows extensive binding between IdeS and Fc regions important of Fc-receptor binding, which contributes to explaining how IdeS can block IgG/FcR interactions.

The overall structure of EndoS-IgG1 Fc indicates that EndoS has the same V-shape that has been shown in previous solved structures of the apo-form. One C γ 2 domain in Fc binds across the ends of the V between the glycosidase domain and the proposed CBM (carbohydrate binding domain). Interestingly, the N-linked glycan that is normally buried between the two C γ 2 domains is flipped out and bound within the glycosidase cavity of EndoS. This is a quite remarkable finding that not only pinpoints EndoS' mode of action, but also substantiates previous work suggesting conformational heterogeneity of the IgG Fc glycan by locking it in this position.

Previous studies have shown evidence for the role of the non-enzymatic domains of the EndoS for interactions with IgG that could explain the protein specificity. One hypothesis has been that the putative CBM in the C-terminal of the enzymes binds the glycans of IgG. However, the crystal structure presented here, shows that the glycosides domain interacts with the Fc glycan, while the CBM interacts with Fc in the Cgamma2-Cgamma3 interface. Furthermore, W803 previously shown to be necessary for hydrolytic activity acts as a hydrophobic plug binding within the cavity at the Cgamma2-Cgamma3 interface. This clearly shows that the CBM provides additional specificity for the IgG Fc protein, but NOT the glycan.

Overall, this study a true tour de force in structural enzyme biology. The findings presented here is of great importance for understanding the molecular details of bacterial immune evasion with implications for development of anti-infective therapy and vaccines. The enzymes have also shown potential as therapy against antibody-mediated autoimmune disease, where molecular details about their activity could allow engineering of enzymes that are not only IgG specific, but also specific for pathogenic antibodies directed to self antigens.

It is very hard to find any flaws at all in the methodology, presentation of results or conclusions. I, however have a few minor suggestions for improvements/clarifications:

Page 3 and references. Ref 14 should not be used to avoid further spread of misleading interpretations of the activity of EndoS hydrolysed IgG. This is a retracted paper due to potential EndoS trace amount in IgG preparations that could account for the effects seen. Part of the dataset, additional experiments, and new interpretations have been published here:

Nandakumar, K.S., M. Collin, K.E. Happonen, S.L. Lundström, A.M. Croxford, B. Xu, R.A. Zubarev, M.J. Rowley, A.M. Blom, C. Kjellman, and R. Holmdahl. Streptococcal Endo- β -N-acetylglucosaminidase suppresses antibody-mediated inflammation in vivo. *Frontiers in immunology*. 2018;9:1623. doi: 10.3389/fimmu.2018.01623. PubMed PMID:30061892; PMCID: PMC6054937.

References: All species names should be in italics (for instance refs 1, 4, 21, 31) and there should be no upper case letters in articles titles.

Reviewer #2 (Remarks to the Author):

Sudol et al Nat. Comm 2022

Sudol et al. engineered a IgG Fc fragment that does not crystallize as readily as conventional IgG Fc's. They then used this Fc fragment to co-crystallize Fc with two *Streptococcus pyogenes* enzymes that modify host IgG's – the protease IdeS and the glycosidase, EndoS; both of these enzymes have medical applications, and have been structurally characterized before as apo or carbohydrate complexes. These structures then rationalize the detailed antibody recognition motifs used by each enzyme. In both instances, the enzyme uses recognition motifs remote from the modification site. I am not deeply conversant with this field, but this strikes me as being an important result whose technical challenges has resulted in it long eluding other researchers.

Overall, this is an interesting paper. The structures are important, and the paper seems well written. My main outstanding concern is the quality of the structures, particularly the EndoS complex structure. Reporting of data collection statistics for this structure is incomplete, but these data seem very weak, and the structure refines with a relatively high Rfree and RSRZ. The authors imply, but do not clearly state, that one EndoS chain is much more poorly ordered. The validation report also shows broken density for the key carbohydrate ligand – though this may reflect the limitations of these reports. Showing a Polder map for this group would resolve this. This is a large complex, so high resolution would be challenging to achieve by crystallography, and most of the structure is anticipated by earlier apo complexes. However, any limitations of the structure need to be presented clearly in the paper if readers are going to be able to trust the results.

In addition, the IdeS crystal is essentially a merohedral twin, which makes interpretation of novel features challenging as the need to detwin the maps during refinement introduces considerable bias towards the model as built. Again, this twinning issue should be stated up front in the results, the challenges this presents to interpreting fine features should be stated, and Polder map showing the binding of the hinge region in the IdeS active site should be presented, if only in the supplementary materials.

Minor issues.

L 46 Is the clinically used EndoS the Mac-1 or the Mac-2 variant?

L 105 Has this trigonal crystal form been reported before for IgG? If so how common is it. Since the orthorhombic crystals only account for half of the known structures, the concern might be that you have only eliminated a portion of the IgG's tendency to self-crystallize.

L 114. What is the evidence for calling this a “less crystallizable variant”? The data presented only gives direct evidence that the two variants crystallize differently. A straightforward way of comparing would be to note the number of conditions within one or more standard 96 well crystal screen(s) that give detectable crystals for each protein variant, and compare these results back to wild type. It might be useful to note any patterns in the conditions that cause crystallization (e.g. if crystallization in high salt conditions is greatly reduced, but the variants continue to crystallize in PEG).

“Uniquely suited” seems a strong claim, since it implies that other mutations with a similar effect are unlikely to exist; perhaps simply “much better suited” or something similar.

L146 – if the authors wish to claim the gel filtration experiments as evidence that IdeS binds IgG as a monomer, then they need to provide calibration of their column. However, the complex is oblate, and SEC is not very precise; SEC MALS or native MS would provide more convincing evidence. Alternatively, the stoichiometry of proteins in the gels presented in Fig S1 could be used to quantify the protein ratio in the complex peak. To my eye though the intensity ratio seems consistent with the Fc band being about 50 % more intense, as would be expected for a 50 kDa and 34 Kda protein forming a 1:1 complex. Possibly the artificially high protein concentrations during SEC help stabilize a non-physiological second IdeS chain binding.

I noticed in fig S1 that the E382A variant seems to convert a considerably larger fraction of the protein into a complex (the complex peak is roughly double the height); this is also the variant for which a structure was reported, also hinting at a possibly more stable complex. Is there any evidence that IdeS binds this IgG variant more tightly than the other variants?

L154 “and thus displays sequence diversity against” Maybe “and thus deviates in sequence from”

Is there any indication of possible interactions with the Fab portions of the antibody with either enzyme? The protease in particular is binding the inter-domain linker, and could also potentially interact with the proximal, structurally conservative portions of the Fab.

L292 “Within the crystal, we observe a 2:1 stoichiometry of EndoS binding to IgG Fc in the complex”. While IgG is conceptually a single object, it still contains two distinct protein chains, which are the normal basis for calculating protein stoichiometry. It might be clearer to say that “Within the crystal, each of the two IgG Fc’s in the complex binds a distinct EndoS molecule, resulting in a complex with 2:2 stoichiometry.” This language around stoichiometry also needs fixing elsewhere.

L 293 “while its counterpart in chain B, although not fully visible in the electron density, appears to be bound to a second EndoS molecule present in the asymmetric unit of the crystal”. This statement is a bit unclear. Do the authors mean that the copy of EndoS bound to chain B is less well resolved in the map, with higher ADPs? Also, it appears that the saccharide is only observed in one endoS chain. If so, please state this clearly.

L341 – accession number for EndoS should be included for easy reference.

L 358 The design of the IgG Fc variants is not adequately described. The exact residue range, the placement and design of the his tag need to be stated. If this construct is reported elsewhere, a reference should suffice.

L392. Please state the twinning operation and the twin fraction refinement converged upon.

S38 – paen should be peak

S80 – data collection statistics need to be reported for the highest resolution shell.

Table S3:

This table needs to report data collection statistics for the highest resolution shell. The pdb validation report gives an $I/\sigma I$ of 0.26 for the highest shell. This seems to indicate that the data is very weak; possibly the resolution should be cut somewhat from the optimistic 3.2 Å reported.

It is physically implausible that the reported water molecules have ADPs around half that of protein. Water molecules depend upon the protein to order them vs the lattice, and are therefore unlikely to be significantly better ordered than the residues they bind. Are the authors certain that these are not ions, or other solvent components.

I also do not understand how this structure has 3 molecules per a.u. Each Fc chain surely counts as a separate molecule?

For the Endo structure, EDS finds a much worse completeness than the depositor (89 % vs 99%). Could the authors please check and resolve this.

Rcryst and Rfree, as well as RSRZ outliers are all fairly poor for this resolution.

I also have some concerns about the map of the carbohydrate. The map in the validation report looks quite broken up, even at 0.7 sigma, while the map in the paper seems much stronger, at 1.1 sigma. I would have more confidence if the authors could show a polder map for this carbohydrate.

Reviewer #3 (Remarks to the Author):

The manuscript entitled “Extensive substrate recognition by the streptococcal antibody-degrading enzymes IdeS and EndoS” by Crispin and coworkers represents a significant triumph in the field of immunoglobulin biology. The description of two novel complexes formed with IgG1 Fc is straightforward and largely appropriate. However, the implications are substantial. These enzymes have engendered substantial interest for current and future therapeutic strategies, and this manuscript presents data that will be essential to design engineered antibodies resistant to these enzymes. Such designs would allow exogenous antibodies to retain functionality while endogenous antibodies were rendered invisible to Fc gamma-receptor mediated immune responses (and potentially complement). Thus, the manuscript is important and appropriate for Nature Communications.

A few aspects should be addressed:

Introduction: “Similarly, multiple domains within EndoS contribute to substrate recognition and catalysis³¹⁻³³, but the collective mechanism of these has not been fully resolved.” The second statement introduces a large goal that is not directly answered by these structures. It would be appropriate to say that the molecular details of substrate recognition remain undefined.

Results: “We conclude that these Fx variants are uniquely suited for screening attempts, as crystallisation of Fc fragments has been rendered less favourable; we subsequently used these variants for screening of enzyme-Fc complexes.’ Because only one variant has been examined at this point in the manuscript, perhaps state that the success of the S variant indicates other substitutions at E382 are expected to decrease Fc crystallization.

Results: “Consequently, the Cg2 domain in chain A is pulled away from chain B; this is reflected in a greater root mean squared deviation between Cas in the Cg2 domains (1.347 Å compared to 0.675 Å in wild-type Fc 3AVE, calculated in ChimeraX³⁷ for residues 237-341) and higher atomic B factors in this domain (Supplementary Fig. 2a).” This sections should be revised. From the presentation it is

unclear to this reader if the RMSD is between Cg2 domains of the same structure or between the two structures? The latter comparison wouldn't make much sense if a simple rotation/translation occurred. If the former, why do differences in individual Cg2s within the same structure relate to Cg2 positions?

Results: "The inability of IdeS to cleave IgG hinge-mimicking peptides³⁰ also indicates an occlusion of the active site in the absence of substrate, especially given the strong potential of hydrogen bonding and hydrophobic interactions observed at the Fc hinge (discussed in the following section). We therefore conclude that this loop is important for IdeS function, specifically in mediating substrate access to the active site." Occlusion is formally one possible explanation among multiple. Do the apo structures show the active site in a conformation capable of accepting substrate peptides? Similar structures but differing activities may represent different motion regimes. Furthermore, the complex was crystallized with an inactivating mutation that may likewise affect conformation.

Results: "A water molecule observed within the active site (Fig. 2c), held in position via hydrogen bonds to L92, G95 and V171 backbone atoms (within IdeS) and the carbonyl oxygen of L235 in the Fc hinge, is well-placed to act as a base catalyst of the emerging covalent tetrahedral intermediate." This conclusion must be supported with electron density data that is not shown, or this statement heavily revised. The resolution is above what is often accepted for high-confidence placement of water molecules. What is the B factor for this water?

Results "Interestingly, although previous work has indicated that it can bind galactose (albeit with low affinity)³², the CBM doesn't bind carbohydrate, and the N- and C-terminal 3 helix bundles, which are homologous to IgG-binding protein A from *Staphylococcus aureus*^{33,53}, don't bind protein." This statement is not strictly supported by the data and should be revised. Perhaps a more specific statement noting that the CBM doesn't bind IgG1 Fc N-glycan residues in the Fc complex and the three helix bundle doesn't interact with the substrate polypeptide.

The supplemental tables should include the % Ramachandran outliers. Obviously, it can be derived from the presented data, but for completeness it is preferable to state these.

REVIEWER COMMENTS

Reviewer #1 (Remarks to the Author):

This study by Sudol et al, attempts to elucidate the structural mechanisms behind antibody glycan hydrolysis and proteolysis by the antibody-specific endoglycosidase EndoS and IgG protease IdeS from Streptococcus pyogenes. These two enzymes are quite unique among glycosidases and proteases since they in contrast to the majority of these very specifically hydrolyze glycan and the protein backbone, respectively, on IgG antibodies. These activities are of major biological and medical significance, since S. pyogenes is a major human pathogen, but maybe even more importantly since both enzymes are used/being developed as drugs conditions such as transplant and/or biotechnological for analysis and modification of antibodies in research and for antibody drug development.

In order to gain further insight into the structural basis for the extreme IgG specificity of both enzymes, this team of researchers attempted to solve the crystal structures of IdeS and EndoS in complex with their IgG Fc substrate. In order to achieve this, they used a very clever approach to first control Fc's propensity to crystallize on its own. By looking at known Fc structures and mutagenesis, IgG1 Fc variants with reduced tendency to self-associate were generated. These were then co-crystallized with enzymatically inactive forms of IdeS and EndoS.

The overall structure of IdeS-IgG1 Fc shows asymmetric binding of IdeS across the Cgamma2 domains predominantly in a monomeric form. Furthermore, a loop between beta strands seven and eight that is flexible in IdeS alone, curls upwards in the co-crystal to accommodate the IgG hinge within the active site cavity. The crystal structure also confirms interaction between the Fc hinge and the active site in IdeS. Previously, exosite binding between IgG and IdeS providing specificity has been hypothesized. The crystal revealed binding of IdeS across both chains of Fc which could explain the earlier observed slower hydrolysis of monomeric Fc. Interestingly, it also shows extensive binding between IdeS and Fc regions important of Fc-receptor binding, which contributes to explaining how IdeS can block IgG/FcR interactions.

The overall structure of EndoS-IgG1 Fc indicates that EndoS has the same V-shape that has been shown in previous solved structures of the apo-form. One Cgamma2 domain in Fc binds across the ends of the V between the glycosidase domain and the proposed CBM (carbohydrate binding domain). Interestingly, the N-linked glycan that is normally buried between the two Cgamma2 domains is flipped out and bound within the glycosidase cavity of EndoS. This is a quite remarkable finding that not only pinpoints EndoS' mode of action, but also substantiates previous work suggesting conformational heterogeneity of the IgG Fc glycan by locking it in this position.

Previous studies have shown evidence for the role of the non-enzymatic domains of the EndoS for interactions with IgG that could explain the protein specificity. One hypothesis has been that the putative CBM in the C-terminal of the enzymes binds the glycans of IgG. However, the crystal structure presented here, shows that the glycosidase domain interacts with the Fc glycan, while the CBM interacts with Fc in the Cgamma2-Cgamma3 interface. Furthermore, W803 previously shown to be necessary for hydrolytic activity acts as a hydrophobic plug binding within the cavity at the Cgamma2-Cgamma3 interface. This clearly shows that the CBM provides additional specificity for the IgG Fc protein, but NOT the glycan.

Overall, this study a true tour de force in structural enzyme biology. The findings presented here is of great importance for understanding the molecular details of bacterial immune evasion with

implications for development of anti-infective therapy and vaccines. The enzymes have also shown potential as therapy against antibody-mediated autoimmune disease, where molecular details about their activity could allow engineering of enzymes that are not only IgG specific, but also specific for pathogenic antibodies directed to self antigens.

It is very hard to find any flaws at all in the methodology, presentation of results or conclusions. I, however have a few minor suggestions for improvements/clarifications:

Page 3 and references. Ref 14 should not be used to avoid further spread of misleading interpretations of the activity of EndoS hydrolysed IgG. This is a retracted paper due to potential EndoS trace amount in IgG preparations that could account for the effects seen. Part of the dataset, additional experiments, and new interpretations have been published here:

*Nandakumar, K.S., M. Collin, K.E. Happonen, S.L. Lundström, A.M. Croxford, B. Xu, R.A. Zubarev, M.J. Rowley, A.M. Blom, C. Kjellman, and R. Holmdahl. Streptococcal Endo- β -N-acetylglucosaminidase suppresses antibody-mediated inflammation in vivo. *Frontiers in immunology*. 2018;9:1623. doi: 10.3389/fimmu.2018.01623. PubMed PMID:30061892; PMCID: PMC6054937.*

We thank the reviewer for noting this mistake; this has been removed and the updated reference cited instead (reference 19 in manuscript – line 810).

References: All species names should be in italics (for instance refs 1, 4, 21, 31) and there should be no upper case letters in articles titles.

We thank the reviewer for noting this; these errors have now been corrected.

Reviewer #2 (Remarks to the Author):

Sudol et al Nat. Comm 2022

*Sudol et al. engineered a IgG Fc fragment that does not crystallize as readily as conventional IgG Fc's. They then used this Fc fragment to co-crystallize Fc with two *Streptococcus pyogenes* enzymes that modify host IgG's – the protease IdeS and the glycosidase, EndoS; both of these enzymes have medical applications, and have been structurally characterized before as apo or carbohydrate complexes. These structures then rationalize the detailed antibody recognition motifs used by each enzyme. In both instances, the enzyme uses recognition motifs remote from the modification site. I am not deeply conversant with this field, but this strikes me as being an important result whose technical challenges has resulted in it long eluding other researchers.*

Overall, this is an interesting paper. The structures are important, and the paper seems well written. My main outstanding concern is the quality of the structures, particularly the EndoS complex structure. Reporting of data collection statistics for this structure is incomplete, but these data seem very weak, and the structure refines with a relatively high Rfree and RSRZ. The authors imply, but do not clearly state, that one EndoS chain is much more poorly ordered. The validation report also shows broken density for the key carbohydrate ligand – though this may reflect the limitations of these reports. Showing a Polder map for this group would resolve this. This is a large complex, so high resolution would be challenging to achieve by crystallography, and most of the structure is anticipated by earlier apo complexes. However, any limitations of the structure need to be presented clearly in the paper if readers are going to be able to trust the results.

We thank the reviewer for rightly noting the limitations of the EndoS-Fc structure. We have truncated the data to 3.45 Å (from 3.2 Å previously) and updated the data collection statistics in Supplementary Table 3 accordingly, including the statistics for the highest resolution shell. We agree that the limitations of the structure could be made clearer, especially regarding the poorly ordered second EndoS molecule. We have consequently updated the text to rectify this as suggested (line 438, line 568). We also include a B-factor distribution and RSRZ distribution for the EndoS-Fc complex (Supplementary Fig. 9) which depicts how the second EndoS molecule is more poorly ordered compared to the rest of the complex. We also gladly take the suggestion of providing a polder map for the N-linked glycan within the EndoS-Fc complex, and include this within the supplementary materials (Supplementary Fig. 8).

In addition, the IdeS crystal is essentially a merohedral twin, which makes interpretation of novel features challenging as the need to detwin the maps during refinement introduces considerable bias towards the model as built. Again, this twinning issue should be stated up front in the results, the challenges this presents to interpreting fine features should be stated, and Polder map showing the binding of the hinge region in the IdeS active site should be presented, if only in the supplementary materials.

We appreciate the definite possibility of model bias in the IdeS-Fc model, given the high twinning fraction present within the data. We gladly take this suggestion from the reviewer and have included a polder map for Fc residues 231-239 (corresponding to the hinge region in chain A, within the IdeS active site) within the supplementary materials. The polder map shows continuous density at a contour level of 3 sigma (Supplementary Fig. 4d), which agrees with the interpretation of substrate binding at this site. This exercise has prompted us to take a more cautious approach in modelling the Fc hinge termini, and we have thus removed one N-terminal residue from each Fc chain; we attach the updated IdeS-Fc .cif coordinate file and wwPDB validation report. We have also stated the twinning fraction in the main text and addressed the potential for model bias, as suggested (line 244).

Minor issues.

L 46 Is the clinically used EndoS the Mac-1 or the Mac-2 variant?

The clinically-used variant of IdeS is Mac-1, which we have now stated within the introduction (line 54).

L 105 Has this trigonal crystal form been reported before for IgG? If so how common is it. Since the orthorhombic crystals only account for half of the known structures, the concern might be that you have only eliminated a portion of the IgG's tendency to self-crystallize.

Looking at human IgG Fc structures currently deposited in the PDB, we note that this space group ($P3_221$) appears to have not been reported previously for a human, apo IgG Fc structure. There are a group of structures reporting this space group (e.g. PDB code 5DVK), although these Fcs have all been crystallised with a small peptide which binds between the Fc $C\gamma 2$ - $C\gamma 3$ interface, in close proximity to E382. We also realised that the previously reported percentage of ~52% was slightly misleading, since we had included this group of structures) that were determined from Fcs crystallised in complex with a peptide binding at the $C\gamma 2$ - $C\gamma 3$ interface. We therefore re-calculated the figure to contain apo Fc structures only, and have included this new figure in the text (60.2 % of 108 apo Fc structures crystallise in $P2_12_12_1$ – line 89).

We acknowledge that we have not fully eliminated the ability of these Fcs to self-crystallise. However, we believe crystallisation has been hindered well enough in order to generate the reported complexes, a point which is a lot clearer with the inclusion of the crystal counting experiments (addressed in the next comment). We have updated the first paragraph of the “Analysis of Fc crystal structures for fc engineering” section (line 83) to state that we are aiming to counteract Fc self-crystallisation (rather than “eliminate”, which could be misleading). We also discuss the reduced crystallisation of the Fc variants observed in the crystal counting experiments, compared to a wild-type Fc (line 134).

L 114. What is the evidence for calling this a “less crystallizable variant”? The data presented only gives direct evidence that the two variants crystallize differently. A straightforward way of comparing would be to note the number of conditions within one or more standard 96 well crystal screen(s) that give detectable crystals for each protein variant, and compare these results back to wild type. It might be useful to note any patterns in the conditions that cause crystallization (e.g. if crystallization in high salt conditions is greatly reduced, but the variants continue to crystallize in PEG).

We thank the reviewer for rightly questioning this and gladly take their suggestion to include this experiment. We have included the results of the crystallisation screens into Figure 1, which nicely illustrates the effect of the E382 mutations on crystallisation. We depict the results of the E382S variant, to match with the crystal structure depicted in Figure 1b, and have included the results for the E382A and E382R variants within the main text (line 141). The new figure is shown below:

We appreciate that within the manuscript, there is only processed data. For editorial assurance, please find an example of the raw experimental process for the crystal counting, below. These experiments were conducted by A. Sudol and verified by M. Crispin and I. Tews.

“Uniquely suited” seems a strong claim, since it implies that other mutations with a similar effect are unlikely to exist; perhaps simply “much better suited” or something similar.

We thank the reviewer for noting this and appreciate that the phrase “uniquely suited” could be misleading. We have altered this sentence so this phrase is not used (line 160).

L146 – if the authors wish to claim the gel filtration experiments as evidence that IdeS binds IgG as a monomer, then they need to provide calibration of their column. However, the complex is oblate, and SEC is not very precise; SEC MALS or native MS would provide more convincing evidence. Alternatively, the stoichiometry of proteins in the gels presented in Fig S1 could be used to quantify the protein ratio in the complex peak. To my eye though the intensity ratio seems consistent with the Fc band being about 50 % more intense, as would be expected for a 50 kDa and 34 kDa protein forming a 1:1 complex. Possibly the artificially high protein concentrations during SEC help stabilize a non-physiological second IdeS chain binding.

We really welcome the constructive criticism regarding the measurement of in-solution stoichiometries. We believe this is not of central importance to the manuscript and have therefore substantially weakened all claims regarding this aspect that rely on the presented supplementary information. Instead, we restrict claims to those only made by observation of the crystal structure and previously published kinetic analysis (line 229). We believe this evidence is sufficient to warrant the included statement regarding the apparent stoichiometry. We still include this figure (Supplementary Fig. 2) to show how purification of the complexes was achieved.

I noticed in fig S1 that the E382A variant seems to convert a considerably larger fraction of the protein into a complex (the complex peak is roughly double the height); this is also the variant for which a structure was reported, also hinting at a possibly more stable complex. Is there any evidence that IdeS binds this IgG variant more tightly than the other variants?

We thank the referee for highlighting this feature of our data. The chromatograms show the relative retention of the isolated proteins and the putative complex (Supplementary Fig. 2). These are, however, not perfectly controlled, as there are stages of sample manipulation between the different runs (mainly sample concentration). We therefore do not have complete confidence in assigning relative abundance to any particular mutation, and have therefore limited speculation on this topic. Despite this limitation, we feel that inclusion of these chromatograms will be of help to researchers pursuing crystallographic analysis of these types of complexes. We’ve added the word “approximately” (regarding the combination of IdeS/Fc in a 1:1 molar ratio) into the figure legend to highlight the likely variability across the data. In addition, residue 382 is not within the IdeS-Fc binding interface, therefore we feel mutations at this residue are unlikely to influence IdeS binding affinity.

L154 “and thus displays sequence diversity against” Maybe “and thus deviates in sequence from”

We thank the reviewer for this recommendation and have altered this sentence as suggested (line 239).

Is there any indication of possible interactions with the Fab portions of the antibody with either enzyme? The protease in particular is binding the inter-domain linker, and could also potentially interact with the proximal, structurally conservative portions of the Fab.

We thank the reviewer for highlighting this. Within the IdeS-Fc crystal structure, electron density is resolvable for the lower hinge region between residues 229-239 (chain A) and 230-239 (chain B). Within IgG1, the upper hinge consists of a further 13 residues (relative to chain A: starting at residue 216), before the globular Fab domain. We envisage that the Fab is present in a flexible state with respect to the Fc, and so any contacts to the Fab portion would be difficult to model. Furthermore, Vincents *et al.* (reference 30 in the manuscript) demonstrated that IdeS cleaves both full-length IgG and its Fc fragment, which suggests the redundancy of the Fab domains in IdeS-Fc binding and cleavage activity (although there is no comparison of relative cleavage rates). However, we cannot formally exclude the possibility that there is some interaction with the Fabs. We have adjusted the text to address this (line 223).

For the EndoS-Fc structure, we observe electron density for residues 238-444 in the Fc chain A and 238-444 in Fc chain B. Dixon *et al.* (reference 32 in the manuscript) previously showed that EndoS cleaves the majority of N-linked glycans from full-length IgG and various Fc fragments, suggesting that the Fab portions of the antibody are unimportant for complex formation. Nevertheless, again we can't exclude the possibility that the Fab regions interact with the enzyme. We have adjusted the main text accordingly to address this (line 418).

L292 “Within the crystal, we observe a 2:1 stoichiometry of EndoS binding to IgG Fc in the complex”. While IgG is conceptually a single object, it still contains two distinct protein chains, which are the normal basis for calculating protein stoichiometry. It might be clearer to say that “Within the crystal, each of the two IgG Fc’s in the complex binds a distinct EndoS molecule, resulting in a complex with 2:2 stoichiometry.” This language around stoichiometry also needs fixing elsewhere.

We gladly take this suggestion from the reviewer and have changed this description of stoichiometry as suggested (line 477, line 519, line 547, line 550).

L 293 “while its counterpart in chain B, although not fully visible in the electron density, appears to be bound to a second EndoS molecule present in the asymmetric unit of the crystal”. This statement is a bit unclear. Do the authors mean that the copy of EndoS bound to chain B is less well resolved in the map, with higher ADPs? Also, it appears that the saccharide is only observed in one endoS chain. If so, please state this clearly.

We agree with the reviewer that enhanced clarity is needed here. The second EndoS molecule (chain D in the structure), bound to Fc chain B, is less well resolved in the density. We have updated the text to state this clearly, both in the introductory paragraph for the EndoS-Fc complex section of the text (line 376), and the “Stoichiometry of EndoS-Fc complex” section (line 506). We also include a B-factor distribution and RSRZ distribution for this complex in Supplementary Fig. 9, as an illustration of this.

We have also updated the text as suggested to clearly state that only one of the Fc N-linked glycans is modelled in the structure (line 379).

L341 – accession number for EndoS should be included for easy reference.

We have now included the genome accession number for EndoS, found in the UniProt page in which the wild-type structure is reported (PDB code 4NUY) (line 577).

L 358 The design of the IgG Fc variants is not adequately described. The exact residue range, the placement and design of the his tag need to be stated. If this construct is reported elsewhere, a reference should suffice.

The IgG1 Fc constructs were expressed using a pFUSE-hIgG1-Fc vector, in which residues 221-447 of human IgG1 are encoded (comprising the hinge region and Fc domain), and have introduced E382A/S/R mutations with site directed mutagenesis. The antibodies are not his-tagged, as they can be affinity purified using a Protein A column. We thank the reviewer for highlighting that this section of the methods may not have been clear, and we have adjusted the text accordingly to explain this in more detail (line 595). We have additionally included the sequences of all the Fc constructs in Supplementary Fig. 11.

L392. Please state the twinning operation and the twin fraction refinement converged upon.

We have stated this in the main text as rightly suggested by the reviewer (line 182, line 670).

S38 – paen should be peak

Thank you for spotting this error; this has now been corrected (Supplementary Fig. 2).

S80 – data collection statistics need to be reported for the highest resolution shell.

We regret the omission of the statistics for the highest resolution shell and have since corrected this mistake; the statistics for the data truncated at 3.45 Å are now reported in Supplementary Table 3.

Table S3:

This table needs to report data collection statistics for the highest resolution shell. The pdb validation report gives an I/sigI of 0.26 for the highest shell. This seems to indicate that the data is very weak; possibly the resolution should be cut somewhat from the optimistic 3.2 Å reported.

We thank the reviewer for noting this and agree that the 3.2 Å previously reported was too optimistic. The data has subsequently been truncated to 3.45 Å, which gives much better data collection statistics for the highest resolution shell (reported in Supplementary Table 3).

It is physically implausible that the reported water molecules have ADPs around half that of protein. Water molecules depend upon the protein to order them vs the lattice, and are therefore unlikely to be significantly better ordered than the residues they bind. Are the authors certain that these are not ions, or other solvent components.

We thank the reviewer for noting this. With re-refinement of our structure against the data truncated at 3.45 Å, we have taken an abundance of caution to model water structure. We considered not modelling any waters at all; however, several very well-ordered waters are observed inside the protein. Their inclusion improves the fit of the model as they provide a structural role, and should therefore be deposited as part of the model. Inclusion of these waters improved the RSRZ score for the model overall.

Within the updated EndoS-Fc model, the water molecules do not have absurdly low individual B factors; the lowest B factor for an individual water is 49 Å², and only three waters have a B factor below 80. We have additionally checked these waters with the check my metal server (<https://cmm.minorlab.org>), which found them to be outliers when modelled as ions. We therefore do not believe any of these waters are ions.

I also do not understand how this structure has 3 molecules per a.u. Each Fc chain surely counts as a separate molecule?

We thank the reviewer for questioning this. The IgG Fc is a covalently-linked obligate homodimer and it is therefore common to refer to this as a single molecule. Structures in the PDB are also typically reported as one molecule/entity comprising two chains (e.g. 3AVE, 2WAH), and so we have followed suit with IgG Fc in our reported structures.

For the Endo structure, EDS finds a much worse completeness than the depositor (89 % vs 99%). Could the authors please check and resolve this.

We thank the reviewer for noting this; assumedly this discrepancy was due to the weak nature of the data up to 3.2 Å. We have re-refined the structure against the data truncated at 3.45 Å, and the validation report now gives a completeness of 100 % for both EDS and depositor.

Rcryst and Rfree, as well as RSRZ outliers are all fairly poor for this resolution.

We thank the reviewer for rightly noting this, and have now addressed the limitations of the structure in more detail within the main text as suggested (line 376).

I also have some concerns about the map of the carbohydrate. The map in the validation report looks quite broken up, even at 0.7 sigma, while the map in the paper seems much stronger, at 1.1 sigma. I would have more confidence if the authors could show a polder map for this carbohydrate.

We thank the reviewer for the excellent suggestion of including a polder map for the putative glycan. We were conscious of the potential for phase bias so, in calculating the map, we used the crystal structure of EndoS devoid of any carbohydrate (PDB code 4NUY). The resulting map shows continuous density for a glycan-shaped entity that matches that reported in the crystal structure of EndoS in complex with the isolated glycan (PDB code 6EN3) and our reported complex. Taken together with the evidence that the glycan can be refined well with our data, we conclude that there is strong evidence that the resulting structure represents the real conformation. We have included the map within the supplementary information (Supplementary Fig. 8), and have adjusted the text to emphasise that the reported data is simply evidence for that “flipped-out” conformation (line 479).

Reviewer #3 (Remarks to the Author):

The manuscript entitled “Extensive substrate recognition by the streptococcal antibody-degrading enzymes IdeS and EndoS” by Crispin and coworkers represents a significant triumph in the field of immunoglobulin biology. The description of two novel complexes formed with IgG1 Fc is straightforward and largely appropriate. However, the implications are substantial. These enzymes have engendered substantial interest for current and future therapeutic strategies, and this

manuscript presents data that will be essential to design engineered antibodies resistant to these enzymes. Such designs would allow exogenous antibodies to retain functionality while endogenous antibodies were rendered invisible to Fc gamma-receptor mediated immune responses (and potentially complement). Thus, the manuscript is important and appropriate for Nature Communications.

A few aspects should be addressed:

Introduction: "Similarly, multiple domains within EndoS contribute to substrate recognition and catalysis³¹⁻³³, but the collective mechanism of these has not been fully resolved." The second statement introduces a large goal that is not directly answered by these structures. It would be appropriate to say that the molecular details of substrate recognition remain undefined.

We thank the reviewer for this suggestion and have since edited this sentence within the text (line 73).

Results: "We conclude that these Fx variants are uniquely suited for screening attempts, as crystallisation of Fc fragments has been rendered less favourable; we subsequently used these variants for screening of enzyme-Fc complexes.' Because only one variant has been examined at this point in the manuscript, perhaps state that the success of the S variant indicates other substitutions at E382 are expected to decrease Fc crystallization.

We gladly take this suggestion and have updated this statement in the text as suggested (line 158).

Results: "Consequently, the Cg2 domain in chain A is pulled away from chain B; this is reflected in a greater root mean squared deviation between Cαs in the Cg2 domains (1.347 Å compared to 0.675 Å in wild-type Fc 3AVE, calculated in ChimeraX37 for residues 237-341) and higher atomic B factors in this domain (Supplementary Fig. 2a)." This sections should be revised. From the presentation it is unclear to this reader if the RMSD is between Cg2 domains of the same structure or between the two structures? The latter comparison wouldn't make much sense if a simple rotation/translation occurred. If the former, why do differences in individual Cg2s within the same structure relate to Cg2 positions?

We thank the reviewer for highlighting this and agree that the statement is confusing. We have removed this sentence and instead simply state that the Cγ2 domains are shifted slightly when superimposed with a wild-type IgG1 Fc (PDB code 3AVE; included within Supplementary Fig. 4), and state the observation of higher B factors for residues within the chain A Cγ2 domain (the Fc chain being cleaved) (line 232).

Results: "The inability of IdeS to cleave IgG hinge-mimicking peptides³⁰ also indicates an occlusion of the active site in the absence of substrate, especially given the strong potential of hydrogen bonding and hydrophobic interactions observed at the Fc hinge (discussed in the following section). We therefore conclude that this loop is important for IdeS function, specifically in mediating substrate access to the active site." Occlusion is formally one possible explanation among multiple. Do the apo structures show the active site in a conformation capable of accepting substrate peptides? Similar structures but differing activities may represent different motion regimes. Furthermore, the complex was crystallized with an inactivating mutation that may likewise affect conformation.

We thank the reviewer for highlighting this. Although the form of IdeS present in the complex structure differs in sequence to the published apo structure of wild-type IdeS (see Supplementary Fig. 5), the two IdeS structures superimpose very well. The catalytic triad residues are also in the same positions (although we can't say if the C94 side chain position is different in apo and

complexed forms), indicating that the inactivating mutation doesn't alter the active site conformation (Supplementary Fig. 5). We suggest the role of this loop is for active site occlusion, given the distinct conformations of this loop present in the three IdeS apo structures (PDB codes 1Y08, 2AVW and 2AU1) which contrasts the conformation of the loop in the complexed form. In addition, IdeS containing a C94A mutation has been shown to retain its ability to inhibit Fc-mediated effector functions (Lei *et al.* – reference 9 in the manuscript), which suggests this mutant maintains native binding properties by conformational mimicry. We have updated the text to address this (line 279).

Results: "A water molecule observed within the active site (Fig. 2c), held in position via hydrogen bonds to L92, G95 and V171 backbone atoms (within IdeS) and the carbonyl oxygen of L235 in the Fc hinge, is well-placed to act as a base catalyst of the emerging covalent tetrahedral intermediate." This conclusion must be supported with electron density data that is not shown, or this statement heavily revised. The resolution is above what is often accepted for high-confidence placement of water molecules. What is the B factor for this water?

We thank the reviewer for highlighting that a detailed discussion of the catalytic mechanism is beyond the scope of the data, and we agree that the statement is too speculative. We have therefore removed the statement regarding the potential role of the water molecule in the active site (line 297).

Results "Interestingly, although previous work has indicated that it can bind galactose (albeit with low affinity)³², the CBM doesn't bind carbohydrate, and the N- and C-terminal 3 helix bundles, which are homologous to IgG-binding protein A from Staphylococcus aureus^{33,53}, don't bind protein." This statement is not strictly supported by the data and should be revised. Perhaps a more specific statement noting that the CBM doesn't bind IgG1 Fc N-glycan residues in the Fc complex and the three helix bundle doesn't interact with the substrate polypeptide.

We gladly take this suggestion from the reviewer and have updated this section of the text accordingly (line 466).

The supplemental tables should include the % Ramachandran outliers. Obviously, it can be derived from the presented data, but for completeness it is preferable to state these.

We thank the reviewer for highlighting this and have added the % outliers in each of the data collection/refinement tables (Supplementary Tables 1, 2 and 3).

REVIEWERS' COMMENTS

Reviewer #2 (Remarks to the Author):

The authors have addressed all of my earlier concerns in a straightforward manner. I have no further concerns about this manuscript moving towards publication.